# Impact of Probiotic Combination in *InR*^[E19]^/TM2 *Drosophila melanogaster* on Longevity, Related Gene Expression, and Intestinal Microbiota: A Preliminary Study

**DOI:** 10.3390/microorganisms8071027

**Published:** 2020-07-11

**Authors:** Shuang Ma, Hao Sun, Weichao Yang, Mingfu Gao, Hui Xu

**Affiliations:** 1Key Laboratory of Pollution Ecology and Environmental Engineering, Institute of Applied Ecology, Chinese Academy of Sciences, 72 Wenhua Road, Shenyang 110016, China; mas969@nenu.edu.cn (S.M.); haos@spaces.ac.cn (H.S.); yangweichao@iae.ac.cn (W.Y.); mingfujun@163.com (M.G.); 2University of Chinese Academy of Sciences, Beijing 100049, China

**Keywords:** insulin receptor (*InR*), probiotic combination, *Drosophila melanogaster*, lifespan, gut bacteria

## Abstract

The insulin receptor (InR) pertains to the insulin receptor family, which plays a key role in the insulin/insulin-like growth factor (IGF)-like signaling (IIS) pathway. Insulin signaling defects may result in the development of metabolic diseases, such as type 2 diabetes, and the *InR* mutant has been suggested to bear insulin signaling deficiency. Numerous studies have reported that probiotics are beneficial for the treatment of diabetes; however, the effect of probiotics on patients with *InR* deficiency has seldom been reported. Therefore, we chose the *InR*^[E19]^/TM2 *Drosophila melanogaster* to investigate. The results indicated that probiotics significantly reduce the mean and median lifespan of *InR*^[E19]^/TM2 *Drosophila* (by 15.56% and 23.82%, respectively), but promote that of wild-type files (by 9.31% and 16.67%, respectively). Significant differences were obtained in the expression of lifespan- and metabolism-related genes, such as *Imp-L2*, *Tor*, and *GstD2*, between the standard diet groups and the probiotics groups. Furthermore, analysis of 16S rDNA via high throughput sequencing revealed that the gut bacterial diversity of *Drosophila* fed with a probiotic combination also differs from that of *Drosophila* fed with a standard diet. In summary, these findings indicate that a probiotic combination indeed affects *InR*^[E19]^/TM2 *Drosophila*, but not all of its impacts are positive.

## 1. Introduction

Intestinal microbiota is a cooperative community that includes trillions of bacteria. Multiple studies suggest that intestinal bacteria can affect physical health, including metabolism, immune regulation, and behavior, by absorbing minerals, synthesizing vitamins, extracting nutrients and energy from food, etc. [1,2,3,4]. Furthermore, microbial composition and its dysbiosis can aggravate several diseases, including diabetes, obesity, Crohn’s disease, and inflammatory bowel disease [5,6].

The World Health Organization defines probiotics as “live microorganisms which when administered in adequate amounts confer a health benefit on the host” [7]. Recently, in order to exert more benefits, some well-known probiotics and prebiotics have been mixed and manufactured into probiotic combination products [8,9]. Clinical studies have demonstrated that some probiotics can regulate the diversity and functional structure of intestinal bacteria [10,11]. One comprehensive review concluded that, in more than 75% of cases, multi-strain combinations are more effective than their individual components [4]. Nowadays, probiotics constitute a multi-billion dollar industry that continues to grow and that is one of the most common food supplements worldwide. For example, probiotics are added to food, such as yogurt and ice cream, as well as cosmetics. Probiotics can also be commercialized as lyophilized pills [12,13].

Despite their popularity, studies over recent decades have shown that not all probiotics are effective in treating diseases, and findings from these studies are often contradictory. Moreover, no probiotic combination has been authorized as a therapeutic agent by important medical regulatory authorities, such as the European Food Safety Authority and the U.S. Food and Drug Administration [14,15,16]. Therefore, this uncertainty requires more credible evidence to elucidate the multifaceted effects of probiotics.

*Drosophila* offers an advantageous model of organisms, because this genus does not require ethical approval and have a relatively low cost [17,18]. In addition, *Drosophila* and humans have a high degree of similarity in terms of molecular and biological signaling, although these pathways in *Drosophila* are somewhat simpler [19,20,21]. *Drosophila melanogaster* is frequently used for studying genetics, development, signal transduction, cell biology, intestinal bacteria, and metabolism. *D. melanogaster* can also be used to identify host–microbe symbiosis candidate mechanisms that are related to pathogen rejection, exogenous organisms, innate immune regulation, diet, and probiotic or probiotic characteristics [22,23,24,25,26,27,28]. *Drosophila* contains an insulin receptor homologue, encoded by the *InR* gene located at position 93E4-5 on the third chromosome, and the receptor protein is strikingly homologous to the human receptor. The *InR* initiates a series of cascade events in the insulin/insulin-like growth factor (IGF)-like signaling (IIS) pathway that play an important role in metabolic functions and diseases. For these reasons, we chose *Drosophila* as a model system in this study.

Evidence shows that probiotics can improve blood glucose parameters by restoring the normal state of intestinal bacteria [29]. Some probiotics and synbiotics have also been reported to reduce the prevalence of metabolic syndrome. In these trials, *Lactobacillus* and *Bifidobacterium* were used, which are the best-known probiotics [30]. On the contrary, conflicting results revealed that probiotics do not have any effect on the parameters of diabetes [31]. *InR* belongs to the insulin receptor family, and *InR* mutant are thought to have insulin signaling defects. The insulin signaling system is considered to be a common link in the pathogenesis of type 2 Diabetes. Therefore, this study aims to clarify the effects of a microbial pharmaceutics on the lifespan, metabolism-related gene expression, and intestinal bacteria of the *InR*^[E19]^/TM2 *Drosophila*, which helps to evaluate the interaction between these probiotics and insulin deficiency. Our findings also suggest that probiotics should be functionally characterized at strain level and the clinical use of relative products should be more individualized.

## 2. Materials and Methods

### 2.1. Probiotic and Drosophila Husbandry

Tested probiotics product Siliankang^TM^ (Hangzhou Grand Biologic Pharmaceutical INC, Hangzhou, China) is an innovative formula containing 4 different strains of probiotic bacteria: *Bifidobacterium infantis*, *Lactobacillus acidophilus**, Enterococcus faecalis*, and *Bacillus cereus*. It is a microbial pharmaceutics which approved by China Food and Drug Administration (Permission Number: S20060010).

The probiotic was supplied from the beginning to the end of this study. *Drosophila melanogaster* (Canton-S wild-type *Drosophila* (64349) and *InR*^[E19]^/TM2 (#9646)) were procured from the Bloomington *Drosophila* Stock Center (Indiana University, Bloomington, IN, USA). *InR*^[E19]^/TM2 is a mutant induced by ethyl methane sulfonate exposure. In all studies, only male *Drosophila* were selected, which were isolated at 8 h following enclosure. The ingredients of the standard cornmeal–sucrose–yeast diet included 9 g soybean meal, 66.825 g cornflour, 30 g sucrose, 0.5 g sodium benzoate, 25 g yeast, 6.5 g agar, and 6.8 mL propionic acid per liter. *D. melanogaster* was grown at 25 °C and 65% humidity with a 12:12 h light–dark cycle.

The probiotic combination was added to the cooled (40 ℃) liquid media, using the appropriate concentration of active bacterial cell of 1.55 × 10^8^ CFU/mL media. Fresh *Drosophila* diets were provided twice a week, and frequent changes of media helped to reduce fungal contamination.

The following four groups were set up: Wild-type (WT) *Drosophila* fed with the probiotic combination (PRO WT), wild-type *Drosophila* fed with the standard diet (CK WT), *InR*^[E19]^/TM2 *Drosophila* fed with the probiotic combination (PRO *InR*^[E19]^/TM2 ), and *InR*^[E19]^/TM2 *Drosophila* fed with the standard diet (CK *InR*^[E19]^/TM2 ).

### 2.2. Lifespan Assay

Survival assays were conducted on newly enclosed male *Drosophila*. A total of 480 *Drosophila* were stochastically and equally divided into four groups, each with six vials. The numbers of dead and escaped *Drosophila* were recorded every day, and fresh diets were given twice a week. The study was concluded at the time of the death of the last *Drosophila*, and the survival curve was plotted versus time. 

### 2.3. Body Weight

On day 10, 20, and 30, the body weight of the *Drosophila* was recorded. *Drosophila* in each group were first anesthetized under CO_2_. Subsequently, the body weight was measured with an ultrasensitive scale (METTLER TOLEDO AL204). Finally, the average body weight in each group was calculated.

### 2.4. Quantitative Real-Time PCR (qRT-PCR)

Specimens were taken from four groups on day 20. Total RNA was isolated from 15–20 *Drosophila*. Total RNA at 1 µg was reverse-transcribed into cDNA with the FastKing First-Strand Synthesis system (Tiangen, KR116-02, Beijing, China) according to the manufacturer’s instructions. QuantiNova SYBR Green I PCR Master Mix (Qiagen, 208052, Hilden, Germany) was selected for the qRT-PCR. The primer sequences are included in Appendix A. Each sample had three technical replicates and three biological replicates, and the final mRNA relative expression dynamics relative to the control were normalized to the housekeeping mRNA that encodes rp49.

### 2.5. High-Throughput Sequencing of 16S rDNA Amplicons 

*Drosophila* from day 20 were rinsed in 50% bleach, 70% ethanol, and sterile PBS before dissection. Intestines were dissected from individuals. Additionally, Malpighian tubules, trachea, and crops were removed. The collected intestines were homogenized, and the QIAGEN DNeasy® Blood and Tissue Kit (Qiagen, 69504, Hilden, Germany) was used to extract genomic DNA. 

Illumina MiSeq sequencing for genomic DNA and sequencing library construction were completed by GENEWIZ (Suzhou, China). VSEARCH (1.9.6) and Silva 132 were used for the alignment of the 16S rDNA reference database and sequence clustering. Operational taxonomic unit (OTU) clustering was conducted for unique sequences, according to 97% similarity. During the clustering process, chimeric sequences were further removed to obtain the representative sequences of OTU. Random sampling was used to calculate the Shannon and Chao1 alpha diversity indices according to the OTU analysis. To compare differences among the groups in the microbial communities, weighted UniFrac analysis was performed. Beta diversity was visualized by the principal coordinate analysis (PCoA) and nonmetric multidimensional scaling (NMDS) methods. linear discriminant analysis (LDA) effect size (LEfSe) is an analytical method for identifying high-dimensional biomarkers and for revealing genomic characteristics (including genes, metabolism, and species classification). LEfSe analysis was conducted by using the online galaxy website. In addition, redundancy analysis (RDA) was conducted based on the Hellinger method. The phylogenetic investigation of communities by the reconstruction of unobserved states (PICRUSt) and Kyoto Encyclopedia of Genes and Genomes (KEGG) orthologues were used to examine the variation of bacterial functional genes. The RAW reads obtained from this study was deposited in the NCBI SRA database under the accession number PRJNA635991.

### 2.6. Statistics

Survival curves were analyzed by GraphPad Prism 8 software (GraphPad Software, La Jolla, CA, USA) and IBM SPSS Statistics (v. 25, IBM SPSS, Chicago, IL, USA). Gene expression was calculated using the 2^ddCT^ method relative to the level of Rp49 and then analyzed by GraphPad Prism 8 software. PICRUSt analysis was conducted by the Galaxy online analysis platform (http://huttenhower.sph.harvard.edu/galaxy/). Metagenomic Profiles (STAMP) software (Dalhousie University, Halifax, NS, Canada) was used to analyze the KEGG orthologues. This study used two-tailed Student’s *t*-tests to compare two samples; asterisks indicate statistical significance, where *p*-values of less than 0.05 were considered significant, i.e., * *p* < 0.05 and ** *p* < 0.01.

## 3. Results

### 3.1. The Probiotic Combination Decreases the Mean and Median Lifespan of Male InR^[E19]^/TM2 D. melanogaster

The probiotic combination significantly decreased the lifespan of the male *InR*^[E19]^/TM2 *Drosophila* (mean 15.56%, median 23.82%; *p* < 0.01); however, it significantly improved that of the WT *Drosophila* (mean 9.31%, median 16.67%; *p* < 0.01) (Figure 1a,b). This finding is supported by the Kaplan–Meier survival curves (Table 1). Simultaneously, the maximum lifespan of both the WT and the *InR*^[E19]^/TM2 *Drosophila* groups was not affected by the probiotic combination *(p >* 0.05). 

### 3.2. Effect of the Probiotic Combination on Expression of Age and Insulin-like Signaling-Related Genes

We observed the differential expression of some key genes between the CK *InR*^[E19]^/TM2 and PRO *InR*^[E19]^/TM2 groups, as well as between the CK WT and PRO WT groups. *Drosophila* Silent Information Regulator 2 (*Sir2*), a lifespan-related gene, was not affected by the probiotic combination in the two types of *Drosophila* (*p* > 0.05). In the *InR*^[E19]^/TM2 and WT groups, the expression of insulin-like peptide (DILP) genes *dilp2* and *dilp5* was not significantly changed (*p* > 0.05) (Figure 2). For genes that related to the IIS pathway: *Drosophila* imaginal morphogenesis protein-late 2 (*Imp-L2*) and *chico* were remarkably downregulated by the probiotic combination in the PRO *InR*^[E19]^/TM2 group (47.57% ± 7.71% and 74.96% ± 11.51%, respectively; *p* < 0.05), and *Imp-L2* was also significantly decreased in the PRO WT group (37.20% ± 19.54%; *p* < 0.05). In the Tor pathway, *Tor* was upregulated in the two probiotic-supplemented groups (WT = 574.74% ± 262.15%, *p* < 0.05; *InR*^[E19]^/TM2 = 571.41% ± 711.84%, *p* < 0.05), while *4E-BP*, *E74B*, and *S6K* were not distinctly changed (Figure 2). *Cat*, *sod2*, *gclc*, and *GstD2* are oxidative stress-related genes, and compared to the standard diet groups, *GstD2* was significantly downregulated in the PRO *InR*^[E19]^/TM2 (76.28% ± 20.26%; *p* < 0.05) and PRO WT (70.66% ± 30.43%; *p* < 0.05) groups (Figure 2). Among the immune-related genes, the expression of *bsk* and *relish* was not affected by the probiotic combination; however, *dro* was remarkably upregulated in the PRO *InR*^[E19]^/TM2 group (287.60% ± 373.34%; *p* < 0.05), while *upd3* was significantly decreased in the PRO WT group (57.05% ± 17.21%; *p* < 0.05) (Figure 2).

### 3.3. Intestinal Bacterial Community Analyses

A total of 2,620,972 sequences were obtained from all samples. Subsequently, 1,248,471 clean sequences were used for downstream analysis. OTU clustering was conducted for unique sequences based on 97% sequence similarity. The dynamic of alpha diversity indices indicated that the dietary probiotic combination significantly influenced and increased the richness in both species and the evenness in the PRO *InR*^[E19]^/TM2 group (Chao1 index: *p* = 0.033; Shannon index: *p* = 0.048) (Figure 3a). In the PRO WT group, the species richness was also increased (Chao1 index: *p* = 0.04; Shannon index: *p* = 0.079) (Figure 3a). These alterations in the diversity of the intestinal bacterial communities were accompanied by significant shifts in intestinal bacterial taxonomic compositions.

Looking across the data of NMDS (non-metric multidimensional scaling), changes of the intestinal bacterial community in *Drosophila* supplemented with the probiotic combination were found. Specifically, the bacterial communities of four different groups were clustered separately from each other (Figure 3b), which was also re-examined by the Adnois analysis (*p* < 0.05). Simultaneously, probiotic treated samples and CK samples (in both WT and *InR*^[E19]^/TM2 group) were obviously distributed alone axis 2, which indicate the driving force of probiotics on *Drosophila* intestinal microbiota was greater than that of genotype (Adnois analysis, *p* < 0.05). Result obtained from the redundancy analysis (RDA) showed that the relationship between the lifespan or body weight and the intestinal bacteria were closer in the probiotic-supplemented than the standard diet-supplemented *Drosophila* (Figure 3c,d).

The LEfSe algorithm was used to calculate the LDA scores and to detect the higher OTU clustering of microbial communities or species with significant differences in the standard diet and the probiotic combination groups. As shown in Figure 4, Proteobacteria was found to play a crucial role in the CK *InR*^[E19]^/TM2 group, while Lactobacillales, Bacilli, Enterococcaceae and Firmicutes were enriched in the PRO *InR*^[E19]^/TM2 group (Figure 4b); Lactobacillaceae, *Lactobacillus* (bacterial genera), *Pseudomonas* (bacterial genera), Pseudomonadaceae, and Micrococcaceae play a crucial role in the CK WT group, and Leuconostocaceae, *Leuconostoc* (bacterial genera), and Enterococcaceae were enriched in the PRO WT group (Figure 4a). The results of PICRUSt and KEGG show how the functional capacity of the intestinal bacterial community developed under the supplement of the probiotic combination. Only two genes, including genes for *Staphylococcus aureus* infection and other glycan degradation, significantly increased in the PRO *InR*^[E19]^/TM2 group compared to the CK *InR*^[E19]^/TM2 group (Figure 4c). No differences were found in the CK WT and PRO WT groups.

## 4. Discussion

The demand for probiotic food and supplements has increased over the past few decades [32,33]. This is the first report to elucidate the effects and underlying mechanisms of a probiotic combination on *Drosophila* with insulin signal deficiency.

The probiotic combination used in this research contained Bifidobacterium infantis, Lactobacillus acidophilus, Enterococcus faecalis, and Bacillus cereus. As well-known probiotics, Lactobacillus, Bifidobacterium, and Enterococcus have been widely used in many kinds of fermented foods. Reports show that these probiotics have health-promoting effects on diabetes-related metabolic diseases [34,35,36,37,38,39,40]. However, the function of bacteria on their host could be affected by the physical and genetic environment of the host, along with other factors [36]. The probiotic combination used in this study did not prove to be beneficial in terms of the maximum lifespan of the two types of Drosophila, namely, the WT and the *InR*^[E19]^/TM2. Nonetheless, the mean and median life of the *InR*^[E19]^/TM2 Drosophila were remarkably decreased in the PRO *InR*^[E19]^/TM2 group, these results are contrary to the WT Drosophila fed with probiotics. Thus, we support the standpoint that probiotics may not have beneficial effects under all conditions.

A comparison of the lifespan and body weight between the experimental and control groups has been long accepted as an important method to examine the physiological effects of substances on animal models. Previous studies have stated that some probiotics can extend the lifespan of experimental worms, mice, and *Drosophila* via regulating multiple lifespan-related biological processes, such as adherence and colonization in the gut of the nematode, phosphorylation of AMPK, inflammation, oxidative stress, metabolic regulation, and energy homeostasis [4,41,42,43]. Therefore, also based on these previous studies, we propose that the decreased median and mean lifespan of *Drosophila* in PRO *InR*^[E19]^/TM2 group could be linked to changes in crucial signaling pathways and in the homeostasis of intestinal bacteria. 

First, several studies have shown that the variation of genes contained in the IIS pathway could lead to an extension of lifespan [44,45]. Moreover, previous works have demonstrated that major pathways, such as the IIS pathway and the interrelated DILPs, the JNK pathway, dFOXO, and the TOR pathway, work together to regulate the lifespan of adult *Drosophila* [46,47,48,49,50,51,52,53].

Similarly to the *Drosophila* insulin receptor (d*INR*), DILPs have also been reported to be able to regulate aging [43,54,55]. In this work, however, the expression of *dilp2* and *dilp5* did not significantly change in the probiotic-supplemented group. This result differs from Susan Westfall’s findings, which showed that probiotics and synbiotics could downregulate *dilp2*, enhance lean body mass, and extend longevity [4]. Several studies have also shown that increased expression of *Imp-L2* could lead to phenotypic modulation, including increased stored lipids, decreased reproductive capacity, and extended lifespan [56,57]. These changes are in accordance with the downregulation of IIS. Moreover, increased *Imp-L2* has been shown to result in the upregulation of *dilp2*, *dilp3*, and *dilp5* mRNA [56]. However, in our work, under the effect of probiotics, the decreased expression of *Imp-L2* did not affect either the transcriptional level of *dilp2* and *dilp5* or the maximum lifespan of *Drosophila*. Despite the fact that the mean and median lifespan of *InR*^[E19]^/TM2 *Drosophila* were affected by the probiotic combination, *Imp-L2* was decreased in both the *InR*^[E19]^/TM2 and the WT *Drosophila*. Therefore, we deduce that the downregulated *Imp-L2* may not be the key factor which affects the mean and median lifespan of *Drosophila*. 

Previous reports have pointed out that restraining the TOR pathway could extend the lifespan of yeast, worms, *Drosophila*, and mice [51,58,59]. In this study, *4E-BP* and *S6K*, key TOR regulators that enhance mitochondrial activity and exert negative feedback on the IIS pathway, respectively [60,61,62], were not affected by the probiotic combination. In contrast, the upregulation of *Tor* was found in both PRO WT and PRO *InR*^[E19]^/TM2 group, while only the PRO *InR*^[E19]^/TM2 group had a reduced median and mean lifespan, the median and mean lifespan in PRO WT group were extended in this study. Therefore, we speculate that the affected *Tor* gene may be a factor that resulted in the decreased median and mean lifespan of the *InR*^[E19]^/TM2 *Drosophila*, but it had no significant effect on the lifespan of WT *Drosophila*. In this context, changes in the TOR signaling pathways may not have been capable to affect the maximum lifespan.

The JNK and FOXO signaling pathways are important genetic determinants of longevity in *Drosophila* [47,48,49,63]. Enhanced JNK activity could prolong the lifespan, and activated dFOXO is necessary for upregulating JNK. Genes related to the JNK/FOXO pathway, e.g., *bsk* and *foxo*, did not show significant variation, indicating that the probiotic combination did not affect these two signaling pathways in *Drosophila*. 

Organisms are exposed to chronic oxidative stress. Accumulated oxidative damage has been reported to result in aging [64]. Cheng Peng et al. and Chunxu Wang et al. demonstrated that increased oxidative resistance induced by apple polyphenols and resveratrol prolongs the mean lifespan of *Drosophila* [65,66]. These findings are consistent with our results—the gene *GstD2,* which is related to oxidative stress, was downregulated in the PRO *InR*^[E19]^/TM2 group, and the mean/median lifespan of these *Drosophila* was reduced. However, the decreased *GstD2* may not be a crucial factor affecting the lifespan of WT *Drosophila* fed with probiotics. Furthermore, it is noteworthy that only the expression of *GstD2*, but not that of *gclc* or *sod2*, was changed by the probiotic combination, although *sod2* is the key gene for oxidation resistance and is intimately associated with lifespan.

Increased expression of *Sir2*, a gene belonging to the sirtuin family of protein acylases, extends the lifespan [63,67,68,69]. Unchanged *Sir2* expression in both the *InR*^[E19]^/TM2 and WT groups in this study indicates that lifespan regulation by the probiotic combination is not controlled by *Sir2*. 

Aging in *Drosophila* has also been associated with immunity [70]. In this study, immune-related genes *dro* and *upd3* were significantly changed in the PRO *InR*^[E19]^/TM2 and PRO WT groups, respectively. Biochemical characterization confirmed Upd as an activator of the JAnus Kinase (JAK)/Signal Transducer and Activator of Transcription (STAT) pathway, and numerous cytokines and growth factors are affected by JAK/STAT-linked intracellular signals [71]. In our study, the increased *dro* and decreased *upd3* indicate that the probiotic combination participates in the regulation of the inflammatory response, and that its application in WT and *InR*^[E19]^/TM2 *Drosophila* yields different results.

The insulin receptor has important biological significance, and the activity of its *Drosophila* homologue, *InR*, is crucial for survival. Probiotics have been shown to downregulate the expression of *InR* and to extend *Drosophila* longevity [43]. In this study, the slight changes in *InR* mRNA levels have different effects on the mean and median lifespan of PRO WT *Drosophila* and PRO *InR*^[E19]^/TM2 *Drosophila*. Interestingly, although several genes tested in this study were negatively regulated or unchanged by the probiotic combination, the gene *chico* in the PRO *InR*^[E19]^/TM2 group was found to be remarkably downregulated. Indeed, David J. Clancy reported that *chico* is able to accelerate aging [72].

Numerous studies have provided evidence that probiotics can impact the gut microorganisms [73,74,75], and our results support these views. The Chao1 and Shannon indices showed that the species richness and the evenness of the intestinal bacteria in the *Drosophila* on a standard diet were different to those on a probiotics-supplemented diet. These results support the conclusions of previous studies, which suggested that dietary probiotic administration can increase gut bacterial diversity [76,77,78]. These changes in species diversity can be considered to be beneficial effects of the probiotic combination on *Drosophila*, because species-rich microorganisms are more resistant to pathogen invasion, more efficient at resource utilization, and make the intestinal ecosystem more stable [79,80]. The results of NMDS showed that the intestinal bacteria were distinctly clustered in the CK *InR*^[E19]^/TM2 and PRO *InR*^[E19]^/TM2 groups, as well as in the CK WT and PRO WT groups. Our RDA results showed that the lifespan and body weight both had a stronger correlation with the enteric microorganisms in the probiotic treatment group than in the standard diet group.

According to the LDA scores, we found that the microbial communities or species that significantly differed between the standard diet group and the probiotic group in the two types of *Drosophila* were not consistent, but the addition of the probiotic combination indeed altered the main intestinal bacteria. Interestingly, Enterococcaceae were remarkably enriched in the *Drosophila* of both the PRO *InR*^[E19]^/TM2 and the PRO WT groups. Enriched levels of Enterobacteriaceae have been linked to weakness in the elderly and inflammation in a mouse model of colitis [81,82]. Therefore, alterations in the intestinal bacteria of *Drosophila* fed with probiotics at day 20 may cause intestinal barrier dysfunction and senescence.

*S. aureus* is a Gram-positive bacterium that is one of the major causes for community-acquired and hospital-acquired infections worldwide [83]. In this study, enhancement of *S. aureus* infection markers in the PRO *InR*^[E19]^/TM2 group indicates the upregulation of gene families associated with infectious diseases, which implies that the use of probiotics may have negative effects. However, the same phenomenon was not found in the PRO WT group. Therefore, we suggest that different strains of *Drosophila* are affected differently by the probiotic combination. Moreover, no other enriched functional genes were observed in this study. This implies that the probiotic combination had no noteworthy effect on the functional composition of the gut microbiomes in WT *Drosophila*.

*Bacillus cereus* is an important cause of food borne infectious diseases and food poisoning, which carrys the enterotoxin genes *nhe*, *hbl*, and *cytK1* [84]. The instructions of Siliankang^TM^ state that, pharmacologically, *B. cereus* improves the intestinal anaerobic conditions, which is beneficial to the colonization of other probiotics. However, toxicology data and contraindications of the tested product were still lacking in the instructions. Data from our study indicate that Siliankang^TM^ shows no toxicity to the wild type *Drosophila*; however it does shorten the lifespan of *InR*^[E19]^/TM2 *Drosophila* (Figure 1a). We also found that inflammatory factor related genes were up-regulated in Siliankang^TM^-treated *InR*^[E19]^/TM2 *Drosophila* (Figure 2). This may indicate that the *B. cereus* shows adverse effects in *InR*^[E19]^/TM2 *Drosophila*; however, we cannot determine the correlation between *B. cereus* and reduced lifespan because of the lack of detailed data. In our follow up study, the interaction between *B. cereus* and abnormal glucose metabolism will be explored through a single strain experiment.

Some of these results could, however, be affected by the inherent limitations of this study. First, there were no significant differences in body weight between the standard diet group and the probiotic group at day 10, 20, and 30. It is possible that, by using more time points to monitor the changes of body weight, some differences can be revealed, potentially accompanied by new findings. Second, the analysis of intestinal bacteria in this study should comprise more time points for an appropriate dynamic comparison. Analyses of samples from day 10 and day 30, in addition to those from day 20 performed here, would elucidate the dynamic variation of intestinal bacteria affected by the probiotic combination.

## 5. Conclusions

In summary, the use of a probiotic combination has both beneficial and harmful effects on *InR*^[E19]^/TM2 *Drosophila*. We found that probiotic administration can increase the gut bacterial diversity of *Drosophila*, while we also observed the phenomenon of a decreased mean and median lifespan in the probiotic combination-supplemented *InR*^[E19]^/TM2 *Drosophila*. These results remind us that the use of probiotics requires more caution and the careful consideration of specific conditions.

## Figures and Tables

**Figure 1 microorganisms-08-01027-f001:**
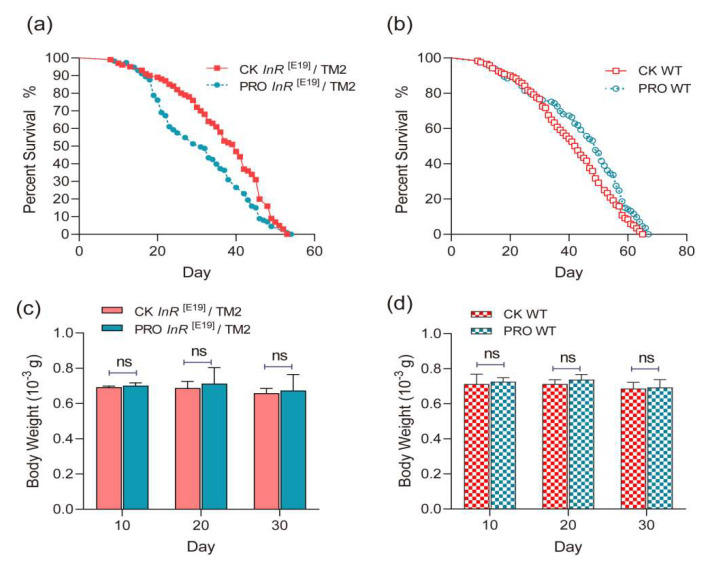
(**a**,**b**) Lifespan curves show how the probiotic treatment impacted the lifespan of the (**a**) *InR*^[E19]^/TM2 and (**b**) wild-type (WT) *Drosophila*. (**c**,**d**) Bar plot shows the body weight dynamics of the (**c**) *InR*^[E19]^/TM2 and (**d**) WT *Drosophila*. No significant difference was observed between the standard diet (CK) and the probiotic treatment in any sample of two groups.

**Figure 2 microorganisms-08-01027-f002:**
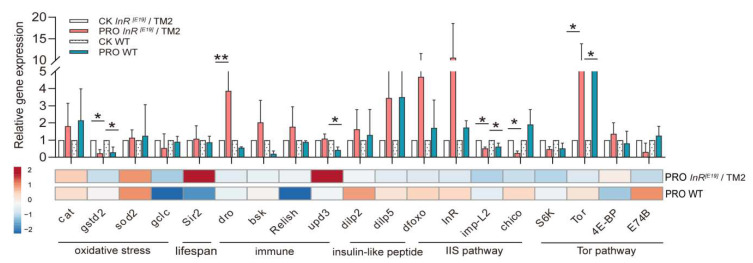
The effect of the probiotic combination on the relative gene transcript levels in WT and *InR*^[E19]^/TM2 *D. melanogaster*. Asterisks indicate a significance between the marked data (* *p* ≤ 0.05 and ** *p* < 0.01). The heat map shows the distribution of the gene differential expression. The color of the squares indicates the normalized relative expression (from 2 to −2), marked on the color block.

**Figure 3 microorganisms-08-01027-f003:**
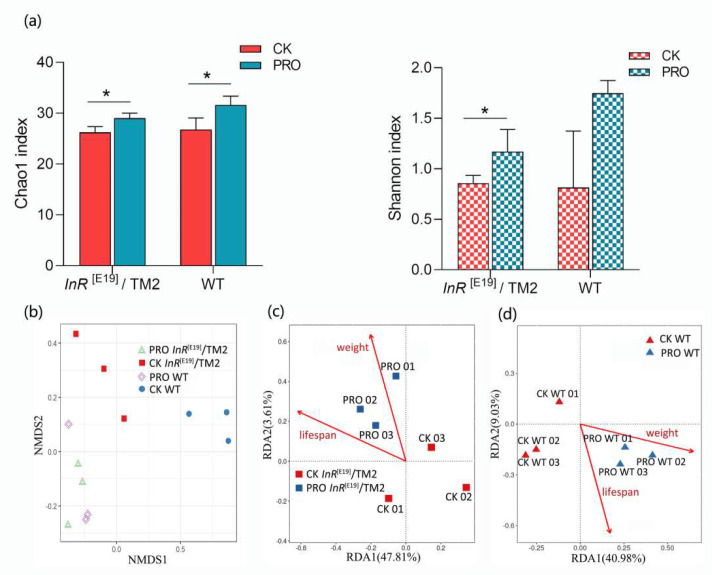
(**a**) Chao1 and Shannon indices indicate intestinal bacterial diversity in all tested samples. Asterisks indicate a significant difference between marked data (* *p* ≤ 0.05). (**b**) NMDS analysis shows the shifts of the bacterial communities driven by the probiotic combination among all samples. (**c**,**d**) Redundancy analysis (RDA) analysis shows the relationship between the bacterial communities, lifespan, bodyweight, and probiotic treatment, (**c**) *InR*^[E19]^/TM2 group, (**d**) WT group.

**Figure 4 microorganisms-08-01027-f004:**
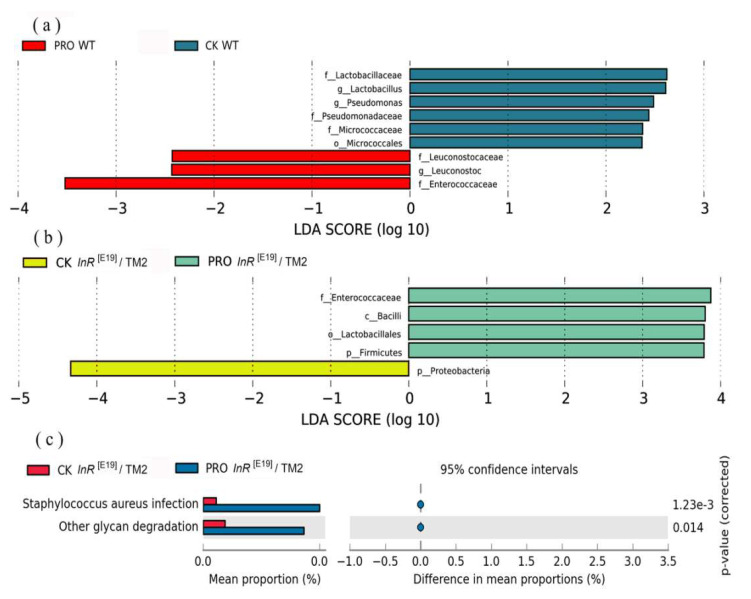
(**a**) Linear discriminant analysis (LDA) scores of the WT *Drosophila* on the probiotics-supplemented diet and the standard (CK) diet. (**b**) LDA scores of the *InR*^[E19]^/TM2 *Drosophila* on the probiotics-supplemented diet and the standard (CK) diet. (**c**) Changes in intestinal bacterial function in the CK *InR*^[E19]^/TM2 and PRO *InR*^[E19]^/TM2 groups (*p* < 0.05).

**Table 1 microorganisms-08-01027-t001:** Statistics for the survival curves. Total number of *Drosophila*, mean and median lifespan (percent change), and log-rank.

-	Total no. of *Drosophila* (n)	Mean (% Change)	Median (% Change)	Log-Rank (vs. Standard Diet)
Standard diet InR	100	36.850	40.238	-
Probiotics InR	113	31.115 (15.56%)	30.652 (23.823%)	p = 0.005335
Standard diet WT	120	40.958333	42.000	-
Probiotics WT	113	44.769912 (9.306%)	49.000 (16.667%)	p = 0.006050

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
