# Peer review of "Impact of Probiotic Combination in InR[E19]/TM2 Drosophila melanogaster on Longevity, Related Gene Expression, and Intestinal Microbiota: A Preliminary Study"

_microorganisms, 2020, doi:10.3390/microorganisms8071027_

Round 1

Reviewer 1 Report

Ma et al. have examined the impact of probiotic administration on lifespan, lifespan associated gene expression and microbiome composition in Drosophila in both a wild-type and insulin signaling mutant. The result is a decrease in the average lifespan of the insulin signaling mutants, with slightly positive effects on the lifespan of the wildtype flies. There were variable impacts on lifespan associated genes and the microbiome in each case with an overall suggestion that the probiotic treatment was detrimental in the insulin signaling mutant. Overall this is an interesting approach to studying the impact of probiotics, but there are some issues that need to be cleared up. Particularly some more information about how the experiments were conducted in order to properly evaluate the results.

Lines 17-18 starts by saying lifespan is not affected, but mean and median lifespan are decreased, contradictory, needs to be fixed

Lines 71-74 again with the contradiction on lifespan. Also did the expression levels of the genes also all decrease? Otherwise this sentence is misleading. For instance Tor increased, not decreased as this sentence implies.

Lines 82-85 More information than just genus level identifications are needed for the probiotics here, this is far too broad, ideally should go to the strain level. This is a crucial detail of the study that needs to be provided.

Line 93-94 I have no idea what this sentence means. What are the “refrigerant” and “concretionary” media? Google searching did not help me either. Is this supposed to be liquid and solid media? And 10^8 to 10^10 CFU/ml is an appropriate concentration of probiotic. Ok, but what was actually used in this study, this is a 100-fold range and could have a big impact on the results.

Lines 110-115 Were these normalized to house keeping genes? Perhaps this is stated in the manufacturer’s handbook, but it is a basic part of the procedure that should be listed. If not, the interpretation of the gene expression results would be brought into question. By the figure, it seems that the probiotic samples are normalized to the normal diet, this should also be described.

Lines 137-144 Statistics for the gene expression?

Lines 147-151 Again, it cannot be said that the lifespan was not affected, but the mean and median decreased. Is this supposed to be interpreted as maximum lifespan was not affected, but the mean and median were? If so, this needs to be more clearly stated. However, this statement also seems to imply that the median and mean decreased in both the mutant and wildtype, but figure 1 and Table 1 only show a decrease in survival for the mutant, the probiotic treated wildtype seem to live slightly longer.

Figure 2 I think this would be better served being displayed a log scale, so that decreases as well as increases can be actually appreciated. Also maybe a bit of explanation as to why we are seeing increases in InR levels in an InR mutant?

Lines 251-254, however, the wild-type increased slightly in mean and median lifespan

Line 319 replace species of flies with strains of flies

Reviewer 2 Report

The major problem with this manuscript is the style and English. There are numerous sites of unclear expressions, wrong English and inconsistencies. These can be found almost in every line of the manuscript. I would recommend to perform extensive editing first.

For example:

Sentences such as: "Some probiotics and synbiotics have also been reported to reduce metabolic syndrome prevalence" need to be checked for correctness.

The authors say: "The results indicated that probiotics combination used in this study not remarkably affected the lifespan of Drosophila, but decreasd the mean and median lifespan." It is not clear whether probiotics do or do not affect lifespan.

Figure legends are either wrong or not sufficient.

Etc.

Round 2

Reviewer 1 Report

The corrections have significantly improved the manuscript, however, they have revealed something else that I think needs to be addressed. Given that Bacillus cereus is one of the components of the probiotic mixture I think there needs to be some discussion of the implications of its inclusion. While so called probiotic strains of B. cereus are used in China there are concerns about how safe these strains actually are. For instance see:

Zhu K, Hölzel CS, Cui Y, Mayer R, Wang Y, Dietrich R, Didier A, Bassitta R, Märtlbauer E, Ding S. Probiotic Bacillus cereus Strains, a Potential Risk for Public Health in China. Front Microbiol. 2016 May 23;7:718. doi: 10.3389/fmicb.2016.00718. PMID: 27242738; PMCID: PMC4876114.

I think given some of the negative effects seen with this probiotic mixture that contains a known foodborne pathogen, this needs to be addressed in the manuscript. 

Reviewer 2 Report

English has been extensively improved, although not completely. There are still some inconsistencies, such asi flora vs. microflora vs. microbiota vs. microorganisms.

In addition, the presentation of figures and figure legends is still insufficient. All figures are doubled, with both parts having the same symbols (a, b, c, d atc), but only one legend. This is unclear. It is also not clear what is the difference between part 1 and 2 of the Figures. There is also inconsistent use of WT vs. Wild atc. Some figures are stretched in width, which makes the presentation unclear (Figure 3, part 1 b, c, d). 
